# Recognition of Early Cardiovascular Disease Symptoms in Hypertensive and Dyslipidemic Individuals of Icheon, Korea: Insights into Educational Levels and Health Literacy

**DOI:** 10.3390/healthcare12070736

**Published:** 2024-03-28

**Authors:** Jeehye Lee, Dong-Hee Ryu

**Affiliations:** 1Department of Preventive Medicine, College of Medicine, Konkuk University, Chungju 27478, Republic of Korea; makemelaugh@kku.ac.kr; 2Department of Preventive Medicine, Daegu Catholic University School of Medicine, Daegu 42472, Republic of Korea

**Keywords:** awareness, cardiovascular diseases, dyslipidemias, education, health literacy, hypertension, myocardial infarction, stroke

## Abstract

The study aimed to explore the relationship between the presence of hypertension or dyslipidemia and the recognition of early symptoms of cardiovascular diseases (CVD), particularly acute myocardial infarction (AMI) and stroke. It is crucial for individuals with hypertension or dyslipidemia to recognize early symptoms of AMI and stroke, as timely and appropriate intervention can lead to favorable health outcomes. The study enrolled 104 participants aged 19 and above who are current residents of the Icheon region, Gyeonggi, Korea. The assessment of early symptoms of AMI and stroke utilized adapted items from the Korea Community Health Survey. In consideration of health literacy and education attainment, logistic regression analyses were conducted. While there was no significant association between hypertension and awareness of AMI or stoke symptoms, individuals with dyslipidemia demonstrated enhanced recognition of specific AMI symptoms, such as ‘sudden chest pain or pressure’ and ‘sudden feeling of breathlessness’. No significant associations were observed between hypertension or dyslipidemia and awareness of stroke symptoms. The study emphasized the significance of targeted health education programs for individuals with chronic conditions to enhance their awareness of early symptoms of AMI and stroke.

## 1. Introduction

Acute myocardial infarction (AMI) and stroke represent significant cardiovascular diseases (CVDs) and are also classified among the three major emergency conditions, alongside severe trauma. According to the 2021 Korean emergency medical statistics, 1.7% of patients experiencing AMI were in critical state upon their arrival at the emergency room, with only 4.1% being promptly discharged [1]. The same data reveal that 13.2% of stroke patients were discharged directly from the emergency room, while 86.2% were either transferred or admitted to the hospital [2]. To mitigate not only the mortality associated with AMI and stroke but also the subsequent complications, it is crucial to emphasize prevention and early intervention when these CVDs manifest.

Lifestyle modification, encompassing strategies like smoking cessation, dietary control, weight management, increased physical activity, and similar measures, is recommended as the first-line approach to prevent adverse CVD outcomes in primary care [3]. However, effective prevention and control necessitate the management of pre-existing conditions such as hypertension and dyslipidemia [4,5]. According to the 2021 Korea National Health and Nutrition Examination Survey, the prevalence of hypertension among Koreans was 21.4% [6]. The 2021 prevalence of hypercholesterolemia was 21.1%, indicating a rising trend [6]. Studies have reported an elevated risk of AMI occurrence in individuals with hypertension [7] and the intra-hospital mortality rate increases when AMI coincides with hypertension [8]. Hypertension heightens not only the risk of AMI but also the occurrence of strokes [9,10]. Furthermore, dyslipidemia is recognized as a key precursor condition for CVD, along with hypertension. Previous studies suggested that dyslipidemia may contribute to the subsequent development of hypertension and vice versa [8,11,12]. Individuals with dyslipidemia exhibit an increased risk of AMI and stroke and display poorer health outcomes [13,14]. Hence, it is crucial for individuals with these pre-existing conditions to recognize early symptoms of AMI and stroke, as timely and appropriate intervention can lead to favorable health outcomes. In addition, the increasing burden of CVD resulting from hypertension and dyslipidemia cannot be overlooked [15,16].

The assessment of knowledge typically entails the formulation of questions and the identification of answers to validate comprehension. This methodology is also applied to the identification of early symptoms associated with CVD. In the Korea Community Health Survey (KCHS), participants are presented with statements pertaining to early emergency symptoms for AMI and strokes [17]. They are instructed to indicate ‘yes’ if they believe the statement represents a relevant early symptom and ‘no’ if they consider it unrelated. The five symptoms indicative of AMI include the ‘sudden onset of pain or tightness in the jaw, neck, or back’, ‘sudden weakness, dizziness, nausea, or cold-sweating’, ‘sudden chest pain or pressure’, ‘sudden pain or discomfort in the arms or shoulders’, and a ‘sudden feeling of breathlessness’ [17]. The five symptoms associated with stroke comprise a ‘sudden loss of strength on one side of your body (face, arm, or leg)’, ‘sudden slurred speech or an inability to understand others’, ‘sudden loss of vision in one eye or double vision’, ‘sudden dizziness or difficulty maintaining balance’, and a ‘sudden severe headache unlike any experienced before’ [17].

It is widely acknowledged that a strong association exists between a lower level of education and poorer health outcomes [18]. A recently highlighted concept is health literacy, defined by the US Institute of Medicine as ‘the degree to which individuals have the capacity to obtain, process, and understand basic health information and services needed to make appropriate health decisions’ [18]. This is recognized as a key mechanism influencing the relationship between educational levels and health outcomes, impacting how well patients comprehend information [18]. Various tools are available for measuring health literacy, with one of the simplest methods being the Single-Item Literacy Screener (SILS) designed for clinical settings [19]. SILS employs a five-point Likert scale to evaluate health literacy, encompassing three questions: ‘How often do you have problems learning about your medical condition because of difficulty understanding written information?’, ‘how confident are you filling out medical forms by yourself?’, and ‘how often do you need someone to help you when you read instructions, pamphlets, or other written materials from you doctor or pharmacist?’ The first question assesses reading comprehension, the second evaluates writing skills, and the third can be used independently to assess the ability to utilize medical information [20].

The study aims to explore the relationship between the presence of hypertension or dyslipidemia and the recognition of early symptoms of CVDs, particularly AMI and stroke. Throughout the investigation, we also examine how this relationship may change in association with educational levels and health literacy.

## 2. Materials and Methods

### 2.1. Study Design and Data Collection

The cross-sectional study was carried out in August 2023 in the Icheon region of Gyeonggi, Korea. The region is comprised of Icheon and Yeoju. The survey was conducted in a self-administered manner; however, in cases when survey participants requested assistance, the researcher provided explanations or additional clarifications for the survey items.

### 2.2. Participants

The study involved individuals aged 19 and above who were residents of the Icheon region and voluntarily indicated their willingness to participate. Quota sampling was implemented, taking into account the age distribution within the region, with the survey striving to achieve a target sample size of 100 participants. Over the survey period, a total of 112 individuals took part, and only those who provided valid responses were considered for analysis. Individuals with missing responses or unreliable answers were excluded, resulting in a total of 104 individuals included in the analysis.

### 2.3. Variables

The items used to assess the recognition of early symptoms associated with AMI and stroke were adapted from the KCHS, as detailed in the introduction. The presence of hypertension or dyslipidemia was defined as indicating either hypertension or dyslipidemia in response to the following question: ‘please indicate all the illnesses you have been diagnosed with by a doctor’. Educational attainment was classified based on the following question: ‘up to what level have you graduated school from?’ with response options of ‘less than elementary school’, ‘elementary school’, ‘junior high school’, ‘high school, ‘college’, and ‘beyond college’. The level of education was dichotomized into ‘less than high school graduation’ and ‘high school graduation or above’ for the analysis. The question of ‘how often do you need someone to help you when you read instruction, pamphlets, or other written materials from your doctor of pharmacist?’ was asked with the following response options: ‘never’, ‘rarely’, ‘sometimes’, ‘often’, ‘and ‘always’. Health literacy was categorized into ‘high (never/rarely)’ and ‘low (sometimes/often/always)’ [19] for the analysis.

### 2.4. Ethical Statement

The study was exclusively conducted with individuals meeting the inclusion criteria. Well-trained researchers, who offered thorough explanations regarding the survey’s purpose and background information, initiated the investigation only after securing written informed consent from eligible participants. Throughout the survey, no personally sensitive information was gathered, and all response data were used solely for analytical purposes. The procedures of this study received approval from the Institutional Review (IRB) at Daegu Catholic Medical Center, and the survey was conducted in accordance with the decisions of the ethics committee (IRB approval number: CR-23-073-L).

### 2.5. Statistical Analysis

Chi-square and Fisher’s exact tests were performed to compare nominal data. Multivariate analyses, evaluating the association between the recognition of early symptoms of CVD and the presence of hypertension or dyslipidemia, were conducted using logistic regression. Three analytic models were designed for the logistic regression: Model 1 (crude), Model 2 (adjusted for the level of education, age, and sex), and Model 3 (adjusted for health literacy, age, and sex). All statistical analyses were performed using SAS version 9.4 (SAS Institute Inc., Cary, NC, USA), and *p*-value < 0.05 was considered to indicate statistical significance.

## 3. Results

The average age of study participants was 51, and the age ranged from 20 to 89 years old. Of all participants, 59.6% were female. While the majority of participants had education beyond junior high school, 47.1% responded that they needed assistance from others when reading documents from health professionals such as doctors or pharmacists. A total of 36 individuals were diagnosed with hypertension and 20 with dyslipidemia by a doctor, and among them 13 individuals had both conditions concurrently (Table 1).

Among the five early symptoms of AMI, the most widely recognized was ‘sudden pain or feeling pressure in the chest’, while awareness of radiating pain was relatively limited. Among the five early symptoms of stroke, speech impairment exhibited the highest level of awareness, and participants demonstrated a fair level of recognition for symptoms such as hemiparalysis, dizziness, imbalance, and headaches. There were no statistically significant differences in the awareness of early symptoms of AMI and stroke according to the presence of hypertension. On the other hand, regarding early symptoms of AMI, there were statistically significant differences in the awareness of ‘sudden chest pain or pressure (*p* = 0.039)’ and ‘sudden feeling of breathlessness (*p* = 0.026)’ according to the presence of dyslipidemia. Other early symptoms of AMI and stroke did not show statistically significant differences according to the presence of dyslipidemia (Table 2).

In the logistic regression, the awareness of early symptoms of AMI did not show statistically significant results according to the presence of hypertension. However, in individuals with dyslipidemia, the likelihood of recognizing ‘sudden chest pain or pressure’ as an early symptom of AMI was 5.13 times higher than the likelihood in those without dyslipidemia when adjusted for sex, age, and health literacy. Likewise, in individuals with dyslipidemia, the likelihood of recognizing ‘sudden feeling of breathlessness’ as an early symptom of AMI was 4.05 times higher than the likelihood in those without dyslipidemia. After further adjustment for sex, age, and health literacy, odds ratio (OR) increased to 4.82 and the association remained statistically significant (Table 3). There were no statistically significant results observed in the recognition of early symptoms of stroke according to the presence of hypertension or dyslipidemia (Table 4).

## 4. Discussion

This study is a cross-sectional investigation conducted with the hypothesis that individuals diagnosed with hypertension or dyslipidemia would exhibit a heightened awareness of early emergency symptoms related to AMI and stroke in the context of educational levels and health literacy.

In a study conducted in the United States, there were no significant differences in the awareness of early symptoms of AMI between individuals with conditions like high blood pressure or elevated cholesterol levels and those without such health issues [21]. Similarly, a study conducted in Korea reported that there was no statistically significant distinction in the awareness of early AMI symptoms between individuals with underlying conditions such as hypertension, diabetes, or dyslipidemia and the non-risk group [22]. The findings of this study indicated no statistically significant association between hypertension and early symptoms of AMI, as the previous studies. However, an association between dyslipidemia and specific early symptoms of AMI, such as ‘sudden chest pain or pressure’ and ‘feeling of breathlessness’, was observed. Such difference may be attributed to the consideration of educational levels and health literacy. Additionally, sudden chest pain or pressure and shortness of breath were reported to have higher awareness compared to other symptoms [23,24]. In a previous study utilizing the 2017 KCHS data, there was substantial evidence supporting increased awareness regarding chest pain or pressure and shortness of breath [22]. The predominant early symptom of AMI was the sudden onset of chest pain, particularly underscored in males, while older individuals commonly identified shortness of breath as an early symptom [24]. However, these studies did not specifically focus on the level of awareness based on the presence of underlying health conditions. The absence of statistically significant differences in awareness levels of other AMI symptoms, irrespective of dyslipidemia presence, may suggest that these symptoms are perceived as atypical, leading to low awareness independent of chronic medical conditions, educational levels, and health literacy. A previous study highlighted that the absence of hyperlipidemia can elevate awareness of atypical AMI presentations [25]. It can be inferred that the difference in the study results may be attributed to the fact that the previous study involved individuals diagnosed with both hypertension and diabetes mellitus. While thrombolytic therapy offers favorable health outcomes in AMI cases, its significance is contingent upon administration within the initial hour of symptom onset [26]. Hence, reducing the time from symptom initiation to hospital arrival is of paramount importance. This underscores the necessity for a targeted health education program to improve the awareness level of early AMI symptoms in detail.

Furthermore, this study exhibited no noteworthy association between the diagnosis of dyslipidemia or hypertension and the awareness of early symptoms associated with stroke. In a previous study, it was reported that a high proportion of the general population recognized hypertension or dyslipidemia as risk factors for stroke [27]. Although there is awareness of risk factors, it is speculated that individuals who actually possess these risk factors may not be more sensitive to early stroke symptoms. Another study also showed that comorbid conditions such as hypertension or dyslipidemia were not associated with decreased time of emergency department presentation [28]. It is noticeable that these results were observed even when considering educational levels and health literacy in this study. It is commonly known that individuals with higher education levels have a higher awareness of stroke risk factors [29]. Previous studies have reported a proportional relationship between health literacy and the recognition of stroke risk factors [30,31]. Taking these data into account, it can be inferred that individuals with chronic conditions, regardless of their educational level or health literacy, may have a similar level of awareness of early stroke symptoms to those without chronic conditions. Targeted consultations or educational programs on early stroke symptoms for clinicians and public health practitioners may be beneficial.

In this study, the awareness rate for early symptoms of AMI (correctly answering all five statements regarding the early symptoms of AMI) was 12.5%, and for stroke it was 36.5%. According to the 2022 KCHS conducted in Icheon, awareness rates for early symptoms of AMI and stroke were 36.3% and 45.6%, respectively [17]. In the same survey for Yeoju in 2022, the awareness rates for early symptoms of AMI and stroke were 42.8% and 51.1%, respectively [17]. The discrepancy between these figures may be attributed to the difference in survey methodology. The KCHS employs the computer-assisted personal interview method and includes a large number of items. Conversely, the present study utilized a concise questionnaire and was conducted with assistance from researchers if needed. Additional research to validate the awareness levels of early symptoms of AMI and stroke as measured by the KCHS is recommended to further elucidate these disparities.

This study has several limitations. First, being a cross-sectional study, it lacks the ability to elucidate causality. Second, despite the quota sampling, findings may be confined to the specific region, leading to limitations in generalizability. In addition, the definition of health literacy may vary from country to country due to population, economy, education conditions, and medical patterns. Third, the presence of hypertension and dyslipidemia was not confirmed through laboratory testing. Fourth, this study deals with a small number of samples in a restricted region of Korea. More comprehensive studies with a large cohort are needed in the future. Nevertheless, the strength of the study is the investigation of the association in context of education and health literacy. Future research could explore the potential factors contributing to the observed patterns, such as cultural or socio-economic influences.

## 5. Conclusions

Targeted educational programs and consultations from clinicians and public health practitioners play a crucial role in improving the recognition of early symptoms of AMI and stroke. In summary, the results of this study have the potential to enhance our comprehension of the factors linked to early symptom awareness for AMI and stroke. Moreover, it is useful work since it covers one of the major problems encountered in the health care system.

## Figures and Tables

**Table 1 healthcare-12-00736-t001:** General characteristics of study participants.

Variables	Subgroups	Total (n = 104)
Age	Mean ± Standard deviation	51.0 ± 18.4
Sex	Male	42 (40.4)
	Female	62 (59.6)
Level of education	<High school	20 (19.2)
	≥High school	84 (80.8)
Health literacy	Low (Sometimes/often/always)	49 (47.1)
	High (never/rarely)	55 (52.9)
Hypertension	Yes	36 (34.6)
Dyslipidemia	Yes	20 (19.2)

Values are represented as number and (%) except for the age.

**Table 2 healthcare-12-00736-t002:** Early symptom recognition according to presence of hypertension and dyslipidemia.

Cardiovascular Disease	Early Symptoms	Total	Hypertension			Dyslipidemia		
			No	Yes	*p*	No	Yes	*p*
Acute myocardial infarction	Sudden onset of pain or tightness in the jaw, neck, or back	29 (27.9)	19 (27.9)	10 (27.8)	0.986	23 (27.4)	6 (30.0)	0.814
	Sudden weakness, dizziness nausea, or cold-sweating	57 (54.8)	38 (55.9)	19 (52.8)	0.762	44 (52.4)	13 (65.0)	0.308
	Sudden chest pain or pressure	74 (71.2)	47 (69.1)	27 (75.0)	0.529	56 (66.7)	18 (90.0)	0.039
	Sudden pain or discomfort in the arms or shoulders	28 (26.9)	18 (26.5)	10 (27.8)	0.886	24 (28.6)	4 (20.0)	0.437
	Sudden feeling of breathlessness	66 (63.5)	41 (60.3)	25 (69.4)	0.357	49 (58.3)	17 (85.0)	0.026
Stroke	Sudden loss of strength on one side of your body (face, arm, or leg)	68 (65.4)	45 (66.2)	23 (63.9)	0.816	52 (61.9)	16 (80.0)	0.126
	Sudden slurred speech or an inability to understand others	74 (71.2)	48 (70.6)	26 (72.2)	0.861	57 (67.9)	17 (85.0)	0.128
	Sudden loss of vision in one eye or double vision	56 (53.9)	35 (51.5)	21 (58.3)	0.504	42 (50.0)	14 (70.0)	0.107
	Sudden dizziness or difficulty maintaining balance	67 (64.4)	45 (66.2)	22 (61.1)	0.608	52 (61.9)	15 (75.0)	0.272
	Sudden severe headache unlike any experienced before	53 (51.0)	35 (51.5)	18 (50.0)	0.887	41 (48.8)	12 (60.0)	0.368

Values are represented as number and (%).

**Table 3 healthcare-12-00736-t003:** Recognition of early symptoms of acute myocardial infarction according to the presence of hypertension or dyslipidemia.

Models	Variables	Sudden Onset of Pain or Tightness in the Jaw, Neck, or Back	Sudden Weakness, Dizziness, Nausea, or Cold-Sweating	Sudden Chest Pain or Pressure	Sudden Pain or Discomfort in the Arms or Shoulders	Sudden Feeling of Breathlessness
HYPERTENSION						
Model 1A	Hypertension (ref = no)	0.99 (0.40–2.44)	0.88 (0.39–1.99)	1.34 (0.54–3.34)	1.07 (0.43–2.65)	1.50 (0.63–3.54)
Model 2A	Hypertension (ref = no)	1.57 (0.55–4.54)	1.07 (0.42–2.69)	0.96 (0.34–2.73)	1.65 (0.57–4.80)	1.19 (0.45–3.16)
	Level of education (ref = ≥high school)	1.67 (0.40–6.96)	0.89 (0.27–2.94)	0.97 (0.24–3.85)	1.22 (0.28–5.36)	0.68 (0.19–2.43)
	Age	0.97 (0.94–1.00)	0.99 (0.96–1.02)	1.02 (0.99–1.05)	0.97 (0.94–1.01)	1.02 (0.99–1.05)
	Sex (ref = male)	1.15 (0.45–2.95)	1.30 (0.56–3.01)	0.57 (0.23–1.48)	0.89 (0.35–2.28)	0.90 (0.38–2.17)
Model 3A	Hypertension (ref = no)	1.54 (0.53–4.46)	1.18 (0.46–3.04)	1.01 (0.35–2.91)	1.65 (0.56–4.83)	1.35 (0.50–3.68)
	Health literacy (ref = high)	1.06 (0.44–2.58)	0.52 (0.23–1.16)	0.71 (0.30–1.71)	0.98 (0.40–2.42)	0.46 (0.20–1.07)
	Age	0.98 (0.95–1.00)	0.99 (0.97–1.02)	1.02 (0.99–1.05)	0.97 (0.95–1.00)	1.01 (0.99–1.04)
	Sex (ref = male)	1.19 (0.46–3.04)	1.44 (0.62–3.39)	0.60 (0.23–1.55)	0.91 (0.35–2.32)	0.98 (0.40–2.38)
DYSLIPIDEMIA						
Model 1B	Dyslipidemia (ref = no)	1.14 (0.39–3.31)	1.69 (0.61–4.65)	4.50 (0.98–20.78)	0.63 (0.19–2.06)	4.05 (1.10–14.88)
Model 2B	Dyslipidemia (ref = no)	1.85 (0.55–6.27)	2.14 (0.71–6.49)	4.54 (0.91–22.72)	0.93 (0.25–3.43)	3.79 (0.95–15.08)
	Level of education (ref = ≥high school)	1.68 (0.40–7.01)	0.92 (0.27–3.09)	1.00 (0.24–4.19)	1.20 (0.27–5.26)	0.69 (0.18–2.59)
	Age	0.97 (0.94–1.00)	0.99 (0.96–1.01)	1.01 (0.98–1.04)	0.98 (0.95–1.01)	1.01 (0.98–1.04)
	Sex (ref = male)	1.01 (0.40–2.56)	1.22 (0.53–2.80)	0.54 (0.21–1.36)	0.81 (0.32–2.02)	0.81 (0.34–1.93)
Model 3B	Dyslipidemia (ref = no)	1.81 (0.53–6.22)	2.55 (0.51–7.99)	5.13 (1.00–26.20)	0.91 (0.24–3.43)	4.82 (1.17–19.84)
	Health literacy (ref = high)	1.04 (0.43–2.54)	0.47 (0.21–1.08)	0.61 (0.25–1.48)	1.05 (0.43–2.57)	0.40 (0.17–0.95)
	Age	0.98 (0.95–1.00)	0.98 (0.96–1.01)	1.01 (0.98–1.03)	0.98 (0.96–1.01)	1.01 (0.98–1.03)
	Sex (ref = male)	1.06 (0.43–2.66)	1.34 (0.58–3.09)	0.56 (0.22–1.42)	0.82 (0.33–2.03)	0.86 (0.36–2.06)

Values are represented as odds ratios and 95% confidence intervals. Abbreviations: ref—reference. Model 1: crude; Model 2: adjusted for level of education, age, and sex; Model 3: adjusted for health literacy, age, and sex; Model A: for hypertension; Model B: for dyslipidemia.

**Table 4 healthcare-12-00736-t004:** Recognition of early symptoms of stroke according to the presence of hypertension or dyslipidemia.

Models	Variables	Sudden Loss of Strength on One Side of Your Body (Face, Arm, or Leg)	Sudden Slurred Speech or Inability to Understand Others	Sudden Loss of Vision in One Eye or Double Vision	Sudden Dizziness or Difficulty Maintaining Balance	Sudden Severe Headache Unlike Any Experienced Before
HYPERTENSION						
Model 1A	Hypertension (ref = no)	0.90 (0.39–2.11)	1.08 (0.44–2.66)	1.32 (0.58–2.98)	0.80 (0.35–1.86)	0.94 (0.42–2.12)
Model 2A	Hypertension (ref = no)	0.66 (0.24–1.76)	0.83 (0.30–2.31)	0.87 (0.34–2.24)	0.66 (0.25–1.75)	0.86 (0.34–2.17)
	Level of education (ref = ≥high school)	1.78 (0.44–7.18)	1.32 (0.32–5.47)	0.96 (0.28–3.27)	1.60 (0.43–5.99)	1.67 (0.50–5.59)
	Age	1.02 (0.99–1.05)	1.01 (0.98–1.05)	1.02 (0.99–1.05)	1.01 (0.98–1.04)	1.00 (0.97–1.03)
	Sex (ref = male)	1.21 (0.51–2.88)	1.00 (0.40–2.50)	0.56 (0.24–1.33)	1.19 (0.50–2.82)	0.95 (0.41–2.18)
Model 3A	Hypertension (ref = no)	0.66 (0.24–1.79)	0.87 (0.31–2.48)	0.90 (0.35–2.33)	0.67 (0.25–1.79)	0.84 (0.33–2.12)
	Health literacy (ref = high)	0.95 (0.41–2.21)	0.67 (0.28–1.62)	0.82 (0.37–1.84)	0.89 (0.39–2.03)	1.18 (0.54–2.60)
	Age	1.02 (0.99–1.05)	1.02 (0.99–1.05)	1.02 (1.00–1.05)	1.01 (0.99–1.04)	1.01 (0.98–1.03)
	Sex (ref = male)	1.28 (0.54–3.08)	1.10 (0.44–2.76)	0.58 (0.24–1.37)	1.27 (0.54–3.03)	0.98 (0.43–2.25)
DYSLIPIDEMIA						
Model 1B	Dyslipidemia (ref = no)	2.46 (0.76–8.02)	2.68 (0.72–9.95)	2.33 (0.82–6.65)	1.85 (0.61–5.57)	1.57 (0.58–4.24)
Model 2B	Dyslipidemia (ref = no)	2.00 (0.56–7.14)	2.32 (0.58–9.36)	2.05 (0.65–6.40)	1.68 (0.51–5.54)	1.60 (0.54–4.75)
	Level of education (ref = ≥high school)	1.85 (0.45–7.56)	1.37 (0.32–5.81)	0.99 (0.29–3.42)	1.64 (0.44–6.21)	1.72 (0.51–5.79)
	Age	1.00 (0.98–1.03)	1.01 (0.98–1.04)	1.01 (0.99–1.04)	1.00 (0.97–1.03)	0.99 (0.97–1.02)
	Sex (ref = male)	1.26 (0.54–2.96)	1.00 (0.41–2.44)	0.55 (0.24–1.29)	1.25 (0.54–2.90)	0.95 (0.42–2.15)
Model 3B	Dyslipidemia (ref = no)	2.01 (0.56–7.18)	2.56 (0.63–10.50)	2.19 (0.69–6.98)	1.71 (0.52–5.68)	1.53 (0.52–4.55)
	Health literacy (ref = high)	0.85 (0.37–1.97)	0.60 (0.25–1.46)	0.74 (0.33–1.68)	0.80 (0.35–1.84)	1.11 (0.50–2.44)
	Age	1.01 (0.99–1.04)	1.01 (0.98–1.03)	1.01 (0.99–1.04)	1.00 (0.98–1.03)	1.00 (0.98–1.02)
	Sex (ref = male)	1.67 (0.58–3.20)	1.09 (0.44–2.68)	0.57 (0.25–1.32)	1.35 (0.58–3.14)	1.00 (0.44–2.24)

Values are represented as odds ratios and 95% confidence intervals. Abbreviations: ref—reference. Model 1: crude; Model 2: adjusted for level of education, age, and sex; Model 3: adjusted for health literacy, age, and sex; Model A: for hypertension; Model B: for dyslipidemia.

## Data Availability

The datasets analyzed during the current study are available from the corresponding author on reasonable request.

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
