# Peer review of "Recognition of Early Cardiovascular Disease Symptoms in Hypertensive and Dyslipidemic Individuals of Icheon, Korea: Insights into Educational Levels and Health Literacy"

_healthcare, 2024, doi:10.3390/healthcare12070736_

Round 1
Reviewer 1 Report
Comments and Suggestions for Authors
REVIEW REPORT FOR THE STUDY “RECOGNITION OF EARLY CARDIOVASCULAR DISEASE SYMPTOMS IN HYPERTENSIVE AND DYSLIPIDEMIC INDIVIDUALS: INSIGHTS INTO EDUCATIONAL LEVELS AND HEALTH LITERACY”
Journal: Healthcare
The paper "Recognition of early cardiovascular disease symptoms in hypertensive and dyslipidemic individuals: Insights into educational levels and health literacy", performs an analysis of the relationship between the presence of hypertension and/or dyslipidemia and the early perception of cardiovascular disease symptoms, in particular stroke and myocardial infarction, in the Inchon region of Korea..
Title and summary. The title and abstract express well the object of study, objectives, and results of the article.
Structure of the article. The contents are well organized and they adhere to the IMRaD structure. It includes a theoretical framework of the research problem but at this point, I suggest the authors incorporate some other bibliographic references that I miss in the text:
Hong M, Kim B, Chang HJ, Kim TH. Incremental health care expenditures associated with hypertension in South Korea. J Hum Hypertens. 2024 Jan 16. doi: 10.1038/s41371-024-00892-8. Epub ahead of print. PMID: 38228761.
GBD. Causes of death collaborators. Global, regional, and national age-sex-specific mortality for 282 causes of death in 195 countries and territories, 1980–2017: a systematic analysis for the Global Burden of Disease Study 2017. Lancet. 2018;392:1736–88. https://doi.org/10.1016/S0140-6736(18)32203-7 . - DOI
Focusing on the opportunity of the study, it must be said that it is useful work since it covers one of the major problems resulting from a health care system.
Materials and methods.
Regarding the material and methods section, the methodology is tailored to the object of study and the objectives and is explained in a transparent manner while it has been validly applied to guarantee the results.
Results.
The results are significant and they are presented in an adequate and understandable way not only through narration but also with self-explained tables and figures that are also well elaborated in terms of presentation. The results justify and relate to the objectives and methods and the results are of sufficient interest.
Discussion.
The discussion appropriately compares the study results with other works, highlighting the main study findings.
Ariyanti R, Besral B. Dyslipidemia Associated with Hypertension Increases the Risks for Coronary Heart Disease: A Case-Control Study in Harapan Kita Hospital, National Cardiovascular Center, Jakarta. J Lipids. 2019 Apr 30;2019:2517013. doi: 10.1155/2019/2517013. PMID: 31183219; PMCID: PMC6515015.
Fuchs FD, Whelton PK. High Blood Pressure and Cardiovascular Disease. Hypertension. 2020 Feb;75(2):285-292. doi: 10.1161/HYPERTENSIONAHA.119.14240. Epub 2019 Dec 23. PMID: 31865786; PMCID: PMC10243231.
Pinho-Gomes AC, Azevedo L, Copland E, Canoy D, Nazarzadeh M, Ramakrishnan R, Berge E, Sundström J, Kotecha D, Woodward M, Teo K, Davis BR, Chalmers J, Pepine CJ, Rahimi K; Blood Pressure Lowering Treatment Trialists’ Collaboration. Blood pressure-lowering treatment for the prevention of cardiovascular events in patients with atrial fibrillation: An individual participant data meta-analysis. PLoS Med. 2021 Jun 1;18(6):e1003599. doi: 10.1371/journal.pmed.1003599. PMID: 34061831; PMCID: PMC8168843.
Bibliography.
The 22.22% of the bibliography cited in the study belongs to the previous five years.
Overall, it is an interesting study and should be considered for publication in Healthcare, once the minor revisions proposed have been resolved.

Author Response
Thank you for your comments and suggestions.
I've made amendments accordingly.
Please check the attached file.
Thank you in advance.

Reviewer 2 Report
Comments and Suggestions for Authors
Dear editor
I read and reviewed with interest the article entitled Recognition of early cardiovascular disease symptoms in hypertensive and dyslipidemic individuals: Insights into educational levels and health literacy by the authors Jeehye Lee, Dong-Hee Ryu
The article explores the relationship between the presence of hypertension or dyslipidemia and the recognition of early symptoms of cardiovascular diseases (CVD), particularly AMI and stroke. through a questionnaire survey of 104 individuals from the population of Gyeonggi, Korea, evaluating their educational level and the degree to which they can recognize specific symptoms of acute myocardial infarction.
In the way in which the information is addressed and the results obtained, the article should fall into the category of communication and not article, because it is practically informative and specific for a very limited group of the local population.
However, in general the article is well written.
In addition, some specific recommendations would be:
In the title I added that the study was in a local population in Korea, because as it is in its present form it would tend to cover the vast majority of the population when that is not the case.
On page 1, line 13, add the meaning before the abbreviation AMI and not after. line 14.
Page 2 line 46, delete in-hospital and instead add intra-hospital
In the introduction he adds information about other pathologies that comprise the metabolic syndrome, such as type II diabetes and obesity, and about weight associated with CVD.
Page 3 results section, the results of table 1 need to be described in detail and referenced in this section.
page 3 line 141 eliminates "of" and adds "and 20 with dyslipidemia."
Age results should be expressed as median with ranges of minimums and maximums, not as mean but less standard error, by type of distribution.
in the figure captions of all tables add "and" between number (%)
The article would benefit from a flow chart to understand how the patients were recruited, where the exclusion and inclusion criteria were placed.
Thank you very much for allowing me to review this article.
Sincerely the reviewer
Comments on the Quality of English LanguageIn general the article is well written.
Author Response

(The authors gave the same response as above.)

Reviewer 3 Report
Comments and Suggestions for Authors
Author Response

(The authors gave the same response as above.)

Reviewer 4 Report
Comments and Suggestions for Authors
Authors presented paper about the relationship between the presence of hypertension or dyslipidemia and the recognition of early symptoms of CVD. The text of the article is generally well written. I have some comments:
1) It is not clear what the "A" and "B" letters in the model numbering mean? Also, Tables 3 and 4 needs further explanation. In particular, what exactly is given numerically?
2) The conclusion should be more specific, containing the main achievement of the research with scientific novelty.
3) Divide the abstract into sub-sections.
Author Response

(The authors gave the same response as above.)

Round 2
Reviewer 3 Report
Comments and Suggestions for Authors
N/A